**Subject Category:**
Biology (whole organism)

environmental science/environmental chemistry

chlorothalonil, *Stenotrophomonas acidaminiphila* BJ1, biodegradation, degradation pathway

**Author for correspondence:**
Qingming Zhang
e-mail: zhangqingminghf@163.com;
qmzhang@qau.edu.cn

# Biotransformation of chlorothalonil by strain *Stenotrophomonas acidaminiphila* BJ1 isolated from farmland soil

Qingming Zhang[1], Hongyu Liu[1], Muhammad Saleem[2] and Caixia Wang[1]

[1]Key Lab of Integrated Crop Pest Management of Shandong Province, College of Plant Health and Medicine, Qingdao Agricultural University, Qingdao 266109, People's Republic of China
[2]Department of Biological Sciences, Alabama State University, Montgomery, AL 36101, USA

QZ, 0000-0002-1673-4565

Chlorothalonil is a widely used fungicide, but the contamination of soil and water environments by this chemical causes potential threats to biodiversity. Given the metabolic potential of soil microorganisms, there is a need for developing microbiological approaches to degrade persistent compounds, such as chlorothalonil, in contaminated sites. Here in this study, we isolated a bacterial strain (namely, BJ1) capable of degrading chlorothalonil from a chlorothalonil-contaminated farmland soil in the Shandong Province, China. Using 16S rDNA gene sequencing, morphological and biological characteristics, we identified the strain BJ1 as *Stenotrophomonas acidaminiphila*. The strain BJ1 uses chlorothalonil as a sole carbon source. At initial concentrations of 50, 100, 200 and 300 mg l$^{-1}$, it degraded 91.5%, 89.4%, 86.5% and 83.5% of chlorothalonil after 96 h of inoculation under optimum conditions (30°C and pH 7.0). Two metabolites, methyl-2,5,6-trichloro-3-cyano-4-methoxy-benzoate and methyl-3-cyano-2,4,5,6-tetrachlorobenzoate, were detected and identified based on HPLC–MS analysis, which suggests that the strain BJ1 metabolized chlorothalonil through the hydroxylation of chloro-group and hydration of cyano-group. The results of this study highlight the great potential for this bacterium to be used in chlorothalonil pollution remediation.

## 1. Introduction

In modern agriculture, pesticides are extensively used to protect plants against pests and pathogens for increasing the crop

production worldwide [1]. However, injudicious use of pesticides results in the contamination of air, soil and water environment. The environmental contamination of pesticides poses significant threats to beneficial organisms, such as earthworms, fish and animals [2]. The chlorothalonil (2,4,5,6-tetrachloroisophthalonitrile, CAS No. 1897-45-6) is the second most popular broad-spectrum fungicide in worldwide agriculture, which is extensively used to control several pathogens of fruits, vegetables and other crops [3,4]. Meanwhile, its use has raised a number of environmental concerns across the globe. The repeated application of chlorothalonil causes serious risks to the lives of aquatic and terrestrial organisms, as well as humans [5–7]. Its exposure to the human body has been linked to cancer, and it is categorized as a potentially carcinogenic compound by US Environmental Protection Agency and the International Agency for Research on Cancer [8]. Chlorothalonil is moderately persistent in soil with a half-life ranging from several days to six months, even up to 1 year after successive application [9–11]. In addition, several studies have reported the residues and metabolic products (metabolites) of chlorothalonil in the soil and water environments [12–14], along with their adverse effects on soil microbial diversity [15].

Several methods are suggested to remove chlorothalonil from contaminated sites that include but are not limited to these: dechlorination, hydrolysis, photocatalysis and biodegradation [9,16,17]. Among these, the biodegradation of chlorothalonil by the soil bacteria has received significant attention, probably because it is a cost-effective and environmentally friendly method [18]. Soil microbes drive the biodegradation of xenobiotic compounds because they can use them as energy and nutrient resources due to their versatile metabolic capabilities [19]. Although some bacterial species, such as *Enterobacter cloacae* TUAH-1, *Paracoccus* sp. XF-3, *Bacillus subtilis* WB800, *Ochrobactrum lupini* TP-D1, *Pseudomonas* CDS-8, etc. are reported to metabolize chlorothalonil [17,20–23], there has always been a great interest in finding new bacterial species that can be used for the bioremediation of chlorothalonil-contaminated soils under local environmental conditions. Given the higher use of chlorothalonil in the regional agriculture, it is necessary to find new bacterial species suitable for removing chlorothalonil from contaminated agricultural sites. Moreover, it is also highly desired to discover bacterial strains that drive the bioremediation process under a broad range of environmental conditions in the context of climate warming [24].

We isolated strain BJ1 from a chlorothalonil-contaminated farmland soil. It can colonize chlorothalonil-contaminated soil and accelerate the degradation of chlorothalonil, as reported in our previous study [25]. However, in this study, we aimed to characterize the degradation pathway of chlorothalonil by the strain BJ1, using HPLC–MS/MS analysis. Moreover, in this study, we also provide more information about the isolation of this strain, in addition to testing the effect of some environmental factors on the biodegradation of chlorothalonil.

# 2. Material and methods

## 2.1. Chemicals and media

Chlorothalonil (97.0%, purity) was obtained from Qingdao Hansen Biologic Science Co., Ltd, Qingdao, China. Unless otherwise stated, all reagents used in this study were of analytical grade. The bacterial growth medium (Luria-Bertani, LB) contained $10.0 \text{ g l}^{-1}$ tryptone, $5.0 \text{ g l}^{-1}$ yeast extract, $1.0 \text{ g l}^{-1}$ NaCl (pH 7.0). The mineral salt medium (MSM) contained $1.0 \text{ g l}^{-1}$ $NH_4NO_3$, $1.5 \text{ g l}^{-1}$ $K_2HPO_4$, $0.5 \text{ g l}^{-1}$ $KH_2PO_4$, $0.5 \text{ g l}^{-1}$ $MgSO_4$, $0.5 \text{ g l}^{-1}$ $(NH_4)_2SO_4$, $0.5 \text{ g l}^{-1}$ NaCl (pH 7.0). The solid agar medium was prepared by adding $16.0 \text{ g l}^{-1}$ agar. The medium was autoclaved at 121°C. The chlorothalonil was dissolved in acetone as a stock solution ($1000 \text{ g l}^{-1}$). To avoid the possible utilization of acetone as carbon by strain BJ1, the different quantity of chlorothalonil stock solution was filter-sterilized and evaporated before adding the MSM.

## 2.2. Isolation and characterization of strain

To isolate chlorothalonil-degrading bacteria, we collected soil from chlorothalonil-contaminated farmland located in Yantai city, Shandong province, China. The enrichment culture technique was employed to isolate chlorothalonil-degrading bacterial strains. Briefly, 10 g of fresh soil was added to 90 ml MSM solution containing $100 \text{ mg l}^{-1}$ chlorothalonil. Then, the mixture-containing Erlenmeyer flask was incubated in a rotary shaker at 160 r.p.m. and 30°C for 7 days. The final enrichment culture was diluted with sterile water and then spread onto the MSM agar plate containing $100 \text{ mg kg}^{-1}$

chlorothalonil. The plates were incubated at 30°C for 48 h. We picked individual colonies based on their morphologies and tested them for chlorothalonil-degrading capabilities. Among these, one strain (named as BJ1) demonstrated the highest degradation ability, and it used chlorothalonil as the sole carbon for its growth.

We determined the taxonomic identity of the stain BJ1 using its morphological, physiological and biochemical properties, as well as 16S rDNA gene sequencing analysis. The morphological features and physiological and biochemical properties of strain BJ1 were analysed according to *Bergey's Manual of Determinative Bacteriology* [26]. The genomic DNA of strain BJ1 was extracted using an EZNA bacterial DNA kit (Omega Bio-Tek, Norcross, GA, USA). We used 27F (5′-AGAGTTTGATCMTGGCTCAG-3′) and 1492R (5′-GGTTACCTTGTTACGACTT-3′) primers to amplify the 16S rDNA gene. The PCR reaction mixture contained: 2.5 µl of 10 × PCR buffer, 1.6 µl of dNTP (2.5 mM), 0.2 µl of rTaqE (5 U µl$^{-1}$), 0.5 µl of each primer, 0.5 µl of DNA template and sterile filtered milli-Q water to 25 µl. The PCR conditions were: denaturation at 94°C for 5 min, followed by 30 cycles at 94°C for 30 s, 55°C for 30 s, 72°C for 90 s, and a final extension at 72°C for 10 min [27]. The PCR product was ligated into the pMD18-T vector (Takara Biotechnology, Dalian, China) and then transformed into *E. coli* DH5a. The PCR product was sequenced by Sangon Biotech Co., Ltd (Shanghai, China). We deposited the gene sequence of 1464 bp in the NCBI GenBank under the accession number JQ247581.1. The relative sequences were analysed by using the basic local alignment search tool (BLAST) (www.ncbi.nlm.nih.gov/BLAST). A phylogenetic tree was built by the neighbour-joining method using MEGA7 software.

## 2.3. Degradation of chlorothalonil and bacterial growth

The strain BJ1 was cultured in LB medium at 30°C and 160 r.p.m. in a rotary shaker until it reached the stationary stage. The bacterial cell pellet was collected by centrifuging cultures at 5000 r.p.m. for 10 min. Then, the cells were washed twice and resuspended in the MSM. The optical density (OD$_{600\,nm}$) of cells was adjusted to approximately 1.0. Then, we added chlorothalonil into the new MSM as sole carbon and energy source. Unless otherwise stated, we inoculated cells suspension at a 5% (v/v) level into a 10 ml glass tube containing 5 ml MSM with 100 mg l$^{-1}$ of chlorothalonil. We also set non-inoculated controls following the same protocol. The mixture-containing tubes were incubated at 30°C and 160 r.p.m. in a rotary shaker. During incubation, samples of the cultures were collected every 12 h and the residual chlorothalonil was determined. The bacterial growth was determined by measuring the optical density (OD$_{600\,nm}$).

To determine the optimum conditions, we investigated the impact of different pH (5.0, 6.0, 7.0, 8.0 and 9.0) and temperature levels (20, 25, 30, 35 and 40°C) on the biodegradation of chlorothalonil by the strain BJ1 after 4 days of incubation. In addition, the effects of different initial chlorothalonil concentrations (50, 100, 200, 300, 400 and 500 mg l$^{-1}$) were also studied under optimum pH and temperature. All samples were set up in triplicate.

The chlorothalonil residue in MSM was extracted with *n*-hexane, dried with anhydrous sodium sulfate and passed through a 0.22 µm syringe filter for instrument analysis. An Agilent 6890 N gas chromatography (GC) system equipped with an electron capture detector and an HP-5 capillary column (30 m × 0.32 mm × 0.25 µm) was used to determine the content of chlorothalonil. The operating conditions of GC were as follows: injector temperature, 250°C; detector temperature, 300°C; oven temperature was initially set at 80°C for 1 min, ramping at 25°C min$^{-1}$ to 260°C and held for 4 min. Nitrogen was used as carrier gas and the injection volume was 1 µl. The recovery rate of chlorothalonil from MSM was 85.6–107.4%.

## 2.4. Determination of degradation products

To detect the metabolites, the strain BJ1 was cultured in 10 ml MSM containing 100 mg l$^{-1}$ of chlorothalonil as a sole carbon source. The 2 ml of sample was extracted after 4 days of incubation with an equal volume of acetonitrile with 5% NaCl and analysed by high-performance liquid chromatography–electrospray ionization–mass spectrometry (HPLC–ESI–MS). The HPLC–ESI–MS (Agilent) conditions were set up as described by Shi *et al.* [23]: column, Agilent Poroshell 120 EC-C18 (75 mm × 2.1 mm × 2.7 µm); column temperature, 35°C; ultraviolet detector, 230 nm; mobile phase, acetonitrile–water (with 0.1% formic acid added); a gradient of 10% (0–1 min) to 95% (1–20 min); and flow rate of 0.3 ml min$^{-1}$. The MS conditions were: mass scanning range (*m/z*), 80–600; mass detector,

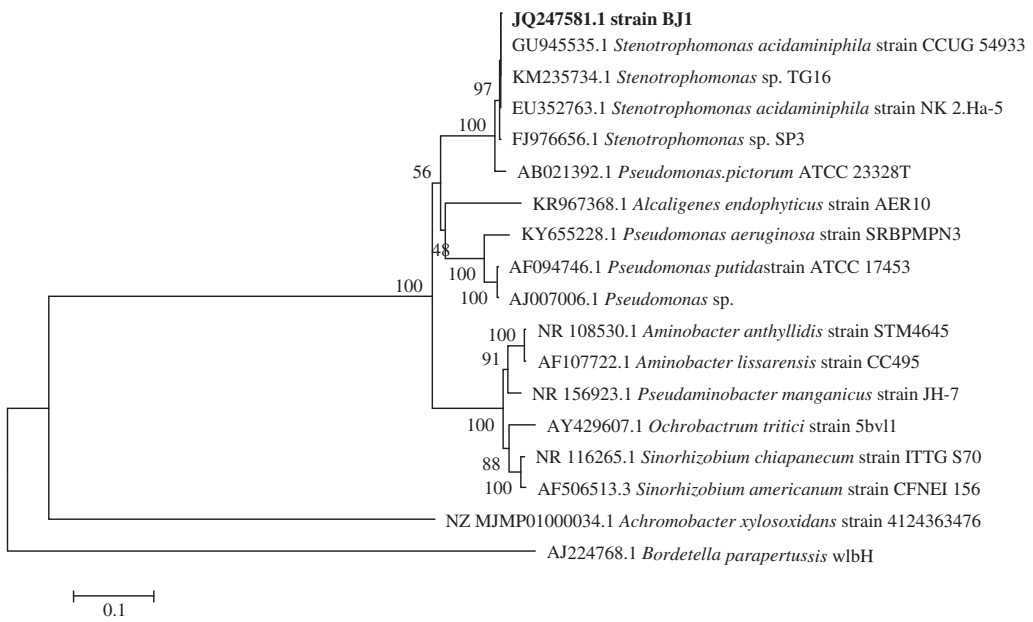

**Figure 1.** Phylogenetic tree of strain BJ1 and related species by the neighbour-joining method using MEGA7. The results were processed by Bootstrap and repeated 1000 times.

quadrupole mass spectrometer; source temperature, 100°C; cone voltage, 30 V; capillary voltage, 3000 V; desolvation gas ($N_2$) flow rate, 500 l h$^{-1}$; and desolvation temperature, 300°C.

## 2.5. Statistical analysis

All statistical analyses were performed in SPSS 18.0 software (SPSS, Chicago, USA). All data are presented as mean ± s.d. One-way ANOVA and Duncan's multiple range tests were used for comparing the differences between treatments at a confidence level of 0.05.

# 3. Results

## 3.1. Isolation and identification of strain BJ1

According to the degradation capacity of chlorothalonil, the strain BJ1 was selected for this study. The colonies of strain BJ1 on LB plate were round with edge neat, uplifted surface, opaque and yellow in colour. This strain was a Gram-negative and asporulate bacterium with a coryneform morphology (0.4–0.5 × 0.2–0.3 μm). Biochemically, the strain was negative for oxidase, urease, nitrate reduction, hydrolysis starch, citrate utilization test and was positive for catalase and gelatin hydrolysis. It did not produce hydrogen sulfide and indole while it was able to use glucose, propionate, malic acid, pyruvic acid and acidic amino acid as sole carbon sources. The strain BJ1 was unable to tolerate 7% sodium chloride. The BLAST results showed that the sequence of 1464 bp of the 16s rDNA gene from the strain BJ1 exhibited 99% similarity with *Stenotrophomonas acidaminiphila* strain CCUG54933, etc. (figure 1); the strain is presented in the same clade in the phylogenetic tree. Based on the biochemical characteristics and BLAST results, the strain BJ1 was named as *S. acidaminiphila* BJ1 (GenBank accession no. JQ247581.1).

## 3.2. Degradation of strain BJ1 on chlorothalonil in MSM

The growth of strain BJ1 and degradation of chlorothalonil (100 mg l$^{-1}$) in the MSM followed the corresponding patterns with time (figure 2). The concentration of chlorothalonil gradually decreased with time, about 97.4% of chlorothalonil was removed in 96 h. Correspondingly, the strain BJ1 growth also reached a plateau in 96 h, and the cell density increased from 0.03 to 0.20 (OD$_{600}$), indicating that this strain could use chlorothalonil as the sole carbon source for its growth. Both temperature and pH influenced the activity of strain BJ1. For instance, the efficiencies of chlorothalonil removal and cell

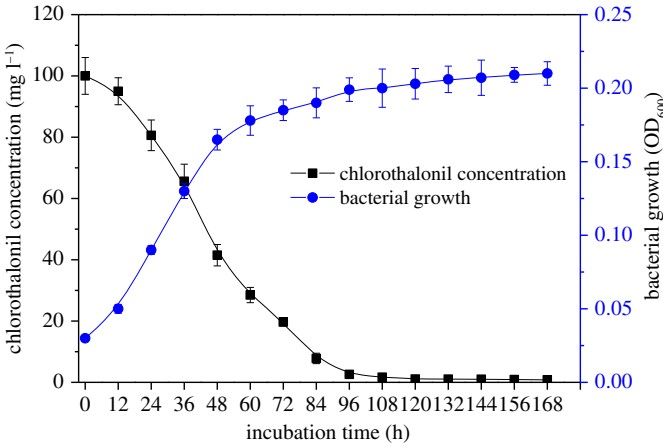

**Figure 2.** Bacterial growth and the corresponding degradation of chlorothalonil under conditions of pH 7.0 and 30℃. Each datum represents the mean of three replicates, and the error bars represent s.d.

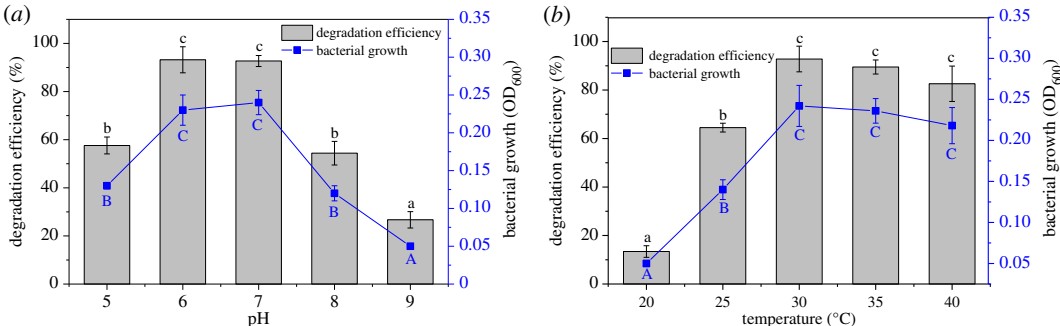

**Figure 3.** Effects of pH and temperature on the degradation of strain BJ1. Each datum represents the mean of three replicates, and the error bars represent s.d. Capital and lower case indicate significantly different at the confidence level of 0.05 between treatments in bacterial growth and degradation efficiency, respectively.

densities at pH 6 and 7 were significantly higher than those of pH 5, 8 and 9. Approximately 93.2% and 92.7% of chlorothalonil were removed at pH 6 and 7, respectively (figure 3a). The relatively lower temperature (20℃ and 25℃) did not accelerate the strain BJ1 growth and thus chlorothalonil degradation. At 30℃, 35℃ and 40℃, the strain BJ1 grew well and demonstrated its ability to remove chlorothalonil efficiently. Among the three temperature treatments, the degradation ability of strain BJI on chlorothalonil was best at 30℃, and the degradation efficiency reached up to 92.8% (figure 3b). Thus, the results of this study indicated that the optimal conditions for BJ1 degradation were pH 6.0–7.0 and 30℃. Under the optimal conditions, strain BJ1 has excellent ability to remove chlorothalonil at the initial concentrations of 50, 100, 200 and 300 mg l$^{-1}$ in MSM, with the degradation efficiencies up to 91.5%, 89.4%, 86.5% and 83.5% at 96 h, respectively (figure 4). However, the removal efficiency of strain BJ1 was not high at higher initial concentrations of chlorothalonil. The strain BJ1 was able to degrade 65.5% and 45.0% of chlorothalonil at initial high concentrations, such as 400 and 500 mg l$^{-1}$ after 96 h, respectively (figure 4).

## 3.3. Degradation products of chlorothalonil by strain BJ1

We detected two chlorothalonil metabolites (peaks A and B) by HPLC after a 4-day incubation (electronic supplementary material, figure S1). Peaks A and B were then detected by the LC–ESI–MS at positive mode (figure 5a,b). According to the mass spectrum, characteristics of fragments ion peaks and previous reports [23,28], product A was identified as methyl-2,5,6-trichloro-3-cyano-4-methoxy-benzoate, and product B was identified as methyl-3-cyano-2,4,5,6-tetrachlorobenzoate. Thus, we suggested that the strain BJ1 metabolized chlorothalonil either through the hydroxylation of chloro-group or hydration of cyano-group, and/or both. However, it is important to mention that the strain BJ1 did not completely mineralize the chlorothalonil.

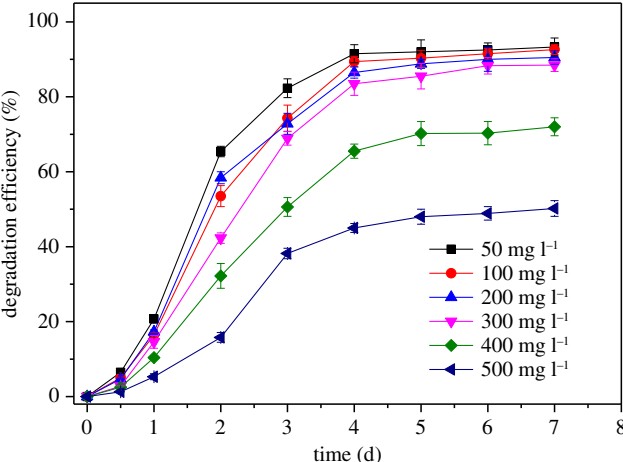

**Figure 4.** Degradation efficiency of strain BJ1 on different initial concentrations of chlorothalonil under conditions of pH 7.0 and 30°C. Each datum represents the mean of three replicates, and the error bars represent s.d.

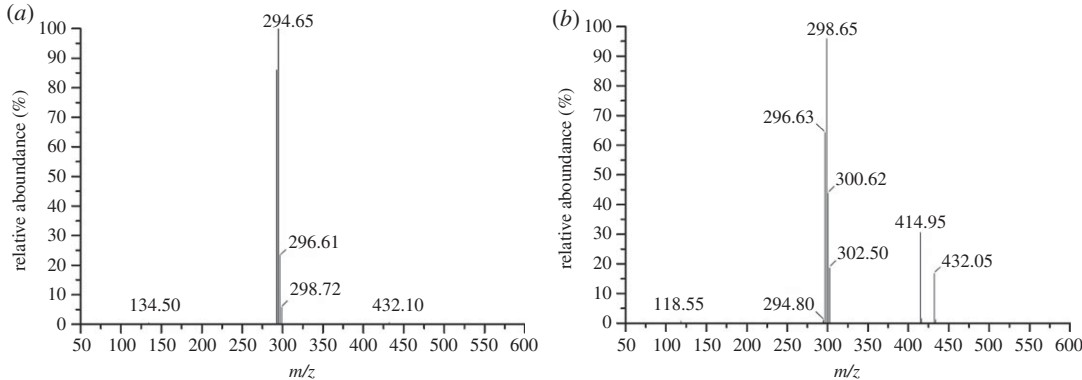

**Figure 5.** Mass spectra of metabolites of chlorothalonil degradation.

## 4. Discussion

Microbial-mediated removal of pesticides has received significant research attention in the last few decades [18,28,29]. With respect to the degradation of chlorothalonil, there are studies that reported some bacterial strains capable of degrading this compound [17,20–23]. In this study, a new highly effective chlorothalonil-degrading strain BJ1 was isolated from a farmland soil by enrichment culture technique. The strain BJ1 is capable of using chlorothalonil as a sole carbon source and could remove approximately 90% of 100 mg l$^{-1}$ chlorothalonil in MSM after 96 h. The degradation ability of strain BJ1 on chlorothalonil is comparable to those of previously reported strains; such as, *E. cloacae* TUAH-1 could remove 97.4% of chlorothalonil (20 mg l$^{-1}$) in aqueous solution after 48 h [20], *Stenotrophomonas* H4 could remove 82.2% of chlorothalonil (20 mg l$^{-1}$) after 7 days incubation in liquid culture [27], and *Ochrobactrum lupini* TP-D1 could degrade 90.4% of chlorothalonil (50 mg l$^{-1}$) after 4 days of incubation in mineral salt broth [23]. The strain BJ1 is identified as *S. acidaminiphila* based on its morphological and biological characteristics as well as the 16S rDNA gene sequence. Previously, bacterial species belonging to the *Stenotrophomonas* genus have been reported to degrade pesticides in the soil and water environments. For example, Dwivedi *et al.* [30] reported *S. acidaminiphila* strain JS-1 that was capable of using butachlor as a sole carbon source, the half-life of butachlor shortened to 4 days in the soil treated with strain JS-1. Moreover, Deng *et al.* [31] reported that *S. acidaminiphila* strain G1 efficiently degraded eight organophosphorus pesticides, such as chlorpyrifos, methyl parathion, methyl paraoxon, diazinon, phoxim, parathion, profenofos and triazophos. According to our knowledge, this is probably the first report that provides detailed information on chlorothalonil degradation by *S. acidaminiphila*.

Both temperature and pH are two important environmental factors that significantly influence the microbial-driven degradation of pesticides in water and soil environments [32,33]. This study also

highlights the significance of temperature and pH on the chlorothalonil degradation by the strain BJ1. Our results indicated that strain BJ1 performed well at a range of temperatures (30–40°C), similar to previously reported results [31]. However, the degradation efficiency of chlorothalonil decreased at the temperature lower than 25°C (figure 3b). It is very likely that low temperatures reduced the bacterial growth and metabolic activity [33,34]. Contrary to this, the strain BJ1 effectively removed chlorothalonil at a higher temperature (40°C) while there are only a few reports on microbial degradation of chemicals at relatively high-temperature conditions [35]. We suggest that this strain is an ideal candidate to develop bioaugmentation strategies to address chlorothalonil contamination particularly in the context of rising soil temperature.

Generally, weak acid and neutral pH (6.0 to 7.0) accelerated the biodegradation of chlorothalonil by strain BJ1, whereas both highly acidic (5) and basic (8, 9) pH conditions were not conducive for chlorothalonil biodegradation (figure 3a), thus suggesting the negative effect of extreme pH conditions on bacterial growth and activity. Our results also suggested that the extreme pH conditions might have negatively influenced the activities of enzymes responsible for the degradation of chlorothalonil [33]; however, further research is needed to confirm this prediction. The initial concentrations of pesticides also affected the degradation efficiency of chlorothalonil by the strain BJ1. Our results indicated that strain BJ1 was able to degrade chlorothalonil over a wide range of initial concentrations (50–500 mg l$^{-1}$). Compared with strain BJ1, the other reported degrading bacteria usually only degrade below 200 mg l$^{-1}$ of chlorothalonil [13,20,23,27], suggesting that train BJ1 has a higher tolerance to chlorothalonil. However, the degradation efficiency of chlorothalonil decreased with increasing concentration, especially in the microcosms spiked with 400 and 500 mg l$^{-1}$ chlorothalonil (figure 4). A similar result was reported by Zhang et al. [27] who found that the degradation efficiency of chlorothalonil decreased as its initial concentration increased in MSM containing degradation bacterium Stenotrophomonas H4. The reason might be due to the toxicity of either chlorothalonil or its degradation products to degrading bacteria [33,36].

Until now, about a dozen of microbial metabolites of chlorothalonil have been reported in previous studies. The main metabolites mainly include hydroxy chlorohydrin, 3-methoxycarbonyl-2,4,5,6-tetrachlorobenzeneacetamide, methyl-2,5,6-trichloro-3-cyano-4-methoxy-benzoate, methyl-3-cyano-2,4,5,6-tetrachlorobenzoate, 4-sulfydryl-2,5,6-trichloroisophthalonitrile, 2-chloro-4,6-disulfydryl-5-chloromethylisophthalonitrile, 2-chloro-4-sulfydryl-5-chloromethylisophthalonitril, etc. [20,23,37]. Here in this study, two degradation products were found in the microcosms after 4 days of incubation. The identified two degradation products are methyl-2,5,6-trichloro-3-cyano-4-methoxy-benzoate and methyl-3-cyano-2,4,5,6-tetrachlorobenzoate. Among these two degradation products, the –COOCH$_3$ group might be esterified in the presence of formic acid during the analysis process [23]. According to previous studies, the precursor compounds of products A and B are 1-carbamoyl-3-cyano-4-hydroxy-2,5,6-trichlorobenzene and 3-cyano-2,4,5,6-tetrachlorobenzamide, respectively [38,39], thus indicating that the chlorothalonil could be degraded by strain BJ1 through conversion of chlorothalonil cyano-group. This proposed degradation pathway is partial to the findings of Shi et al. [23], who also reported the conversion of cyano-group during the bacterial (O. lupini TP-D1) degradation of chlorothalonil. In this regard, some key enzymes in bacteria, such as dehalogenase and glutathione S-transferase, play important roles in the degradation of chlorothalonil [40,41]. Therefore, further research is necessary to study the role of enzymes of the strain BJ1 to understanding the mechanisms underlying the degradation of chlorothalonil. Moreover, it is important to mention that the toxicity of two identified metabolic products has not been reported before. However, in our previous study, we performed a micronucleus test of Vicia faba root tips and found that the genotoxicity of chlorothalonil in the soil was reduced when it was degraded by the strain BJ1 [25]. Thus, our previous and current findings suggest that the strain BJ1 can decrease the toxicity of chlorothalonil. However, further research is required to test the toxicity of the two metabolic products on some soil non-target organisms (e.g. earthworms) [42].

In conclusion, we report the isolation and characterization of a novel chlorothalonil-degrading strain S. acidaminiphila BJ1. The strain efficiently removed chlorothalonil at a wide range of temperature, pH and initial concentration. One of the important aspects of this study is that the strain BJ1 was capable of removing chlorothalonil at 40°C, thus suggesting the performance of this bacteria under climate warming. Although we identified two degradation products and two possible chlorothalonil-degrading pathways, more work is required to underpin the toxicity and metabolic pathway of chlorothalonil by the strain BJ1.

Data accessibility. Data available from the Dryad Digital Repository: https://doi.org/10.5061/dryad.v15fc6b [43]. The figure not in the article has been uploaded as part of the electronic supplementary material.

Authors' contributions. Q.Z. and C.W. conceived and designed the research; Q.Z and H.L. performed the experiment; Q.Z., M.S. and H.L. analysed the data and wrote the paper. All authors gave final approval for publication.

Competing interests. The authors declare no competing interests.

Funding. This work was supported by grants from the Natural Science Foundation (grant no. ZR2016CM11) and Primary Research & Development Plan (grant no. 2017GSF21112) of Shandong Province, China.

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
