## [Reviewer comments · Royal Society Open Science]

Review History

RSOS-190562.R0 (Original submission)

Review form: Reviewer 1

Is the manuscript scientifically sound in its present form?

No

Are the interpretations and conclusions justified by the results?

No

Is the language acceptable?

Yes

Is it clear how to access all supporting data?

No

Do you have any ethical concerns with this paper?

No

Have you any concerns about statistical analyses in this paper?

No

Recommendation?

Major revision is needed (please make suggestions in comments)

Comments to the Author(s)

Manuscript ID RSOS-190562

Title: Biodegradation of chlorothalonil by strain *Stenotrophomonas acidaminiphila* BJ1 isolated from farmland soil.

The manuscript is well organized and written. The manuscript described the degradation potential, in culture media, of a bacteria strain isolated from a contaminated soil in China to remove chlorothalonil under different concentrations and temperatures. The novelty of this article is scarce since there are several manuscripts published and even the main author has already published the degradation potential of this strain in soil under greenhouse conditions, the enzymes implicated, the degradation pathways as well as genotoxicity of chlorothalonil using micronucleus assay of *V. faba*. It is interesting the study related to identify the isolated strain and biodegradation potential under different the environments. In the experimental design no controls were run in parallel with the experimental solutions. The analysis of the metabolites (formation and elimination) during the degradation of chlorotalonyl in the liquid media is important to describe the degradation pathway. Author only detected two metabolites of the proposed pathway and those were not confirmed. The discussion is a bit speculative. The English grammar was not revised. The article must be improved to be published in this journal. Some more comments.

Introduction

The literature reported must be actualized. There are more recent manuscripts related with the subject of this article, pathways, etc... that are not mentioned.

Page 3 line 33 (14) "significant threats to beneficial micro-and microorganisms, including plants". It mean "macro- and microorganisms"???. Please check

Material and method

Describe the biological assays and insert the reference of the biochemical method applied to identify the strain.

Results

Page 9 section 3.3. If possible describe if the two compounds detected were formed at the same time and if were in the same amount or one was relatively more abundant. It is useful to support the degradation pathway. Author must confirm the identity of these compounds.

Discussion

This section must be improved taking into account the different degradation routes described for chlorothalonil in the literature.

Page 10 lines 183-186 (4-12). This information is a repetition and can be omitted.

Page 10 line 200 (48). "According to our knowledge, this is the first study in which we provide information about the metabolic pathway of chlorothalonil degradation by the *S. acidaminiphila* strain". The compounds were not confirmed by the authors. The pathway is a speculation with data from literature and it does not improve the actual knowledge about the biodegradation pathway of chlorothalonil. This sentence must be take with caution.

Page 12 lines 230-23 (12-15). "Among these two degradation products, the -COOCH₃ group might be esterified in the presence of formic acid during the analysis process. This is unclear because in the material and method section the analysis process was omitted.

I suggest read the following reference to be inserted in the discussion section: Chlorothalonil degradation by *Ochrobactrum lupini* strain TP-D1 and identification of its metabolites by Xiu-

Zhen Shi, Rong-Jun Guo, Kazuhiro Takagi, Zuo-Qing Miao, Shi-Dong Li. World J Microbiol Biotechnol (2011) 27:1755–1764. DOI 10.1007/s11274-010-0631-0.

Figures

Figure 1. Please increase the size

Figure 6 can be omitted.

Review form: Reviewer 2

Is the manuscript scientifically sound in its present form?

Yes

Are the interpretations and conclusions justified by the results?

Yes

Is the language acceptable?

Yes

Is it clear how to access all supporting data?

Not Applicable

Do you have any ethical concerns with this paper?

No

Have you any concerns about statistical analyses in this paper?

No

Recommendation?

Major revision is needed (please make suggestions in comments)

Comments to the Author(s)

The work of Zhang et al describes the isolation of a soil bacterium able to biotransform the fungicide chlorothalonil. The authors studied the effect of some conditions (pH, temperature, initial concentration) on the biotransformation of chlorothalonil by this strain and also detected two metabolites. Overall it is a very straight-forward work and I would recommend it for publication after some issues are addressed.

In your 2017 paper (doi: 10.1186/s13568-017-0530-y) you already mention strain BJ1 and have already identified it as *Stenotrophomonas acidaminiphila*, a strain able to utilize chlorothalonil (“The *Stenotrophomonas acidaminiphila* probiotic strain BJ1 was isolated in our laboratory and deposited in China Center for Type Culture Collection”). But this work is presented in the current manuscript as something new. If it is not, then paragraphs 2.2 and 3.1 should be omitted. But if they are indeed omitted, then this manuscript will lack its current completeness.

In the title “biodegradation” could be replaced by “biotransformation.”

Line 177: if the bacterium was not able to mineralize the pollutant, then how was it able to grow with the pollutant as a sole carbon and energy source?

Line 69: do you mean 10 g/L NaCl?

Line 72: what was the concentration of the chlorothalonil stock solution? In which solvent it was dissolved?

Line 82: how did you measure the "chlorothalonil-degrading capabilities"?

Line 104: did you use a glass tube? A polypropylene falcon tube?

Line 109: goal was to determine the "optimum conditions" of bacterial growth/ chlorothalonil degradation/both?

Line 155: a calibration curve to correlate the OD with the dry cell weight is needed. Also in line 103 you mention that the initial OD is 1, but here you say that it was 0.03. elaborate?

Line 222: other microorganisms can tolerate higher concentrations of this pollutant fungicide, compared to BJ1?

Line 226: describe the products and pathways.

Figure legends 2 and 4: state the conditions (pH, temp etc)

Figure 3 could be split into two separate figures (3a and 3b) and the time course be presented.

Figure 6: how certain can you be about the position at which the molecule is transformed? (the Cl at position 4 and CN at position 1).

Decision letter (RSOS-190562.R0)

19-Jul-2019

Dear Dr Zhang,

The editors assigned to your paper ("Biodegradation of chlorothalonil by strain *Stenotrophomonas acidaminiphila* BJ1 isolated from farmland soil") have now received comments from reviewers. We would like you to revise your paper in accordance with the referee and Associate Editor suggestions which can be found below (not including confidential reports to the Editor). Please note this decision does not guarantee eventual acceptance.

Please submit a copy of your revised paper before 11-Aug-2019. Please note that the revision deadline will expire at 00.00am on this date. If we do not hear from you within this time then it will be assumed that the paper has been withdrawn. In exceptional circumstances, extensions may be possible if agreed with the Editorial Office in advance. We do not allow multiple rounds of revision so we urge you to make every effort to fully address all of the comments at this stage. If deemed necessary by the Editors, your manuscript will be sent back to one or more of the original reviewers for assessment. If the original reviewers are not available, we may invite new reviewers.

- Data accessibility

If you wish to submit your supporting data or code to Dryad (<http://datadryad.org/>), or modify your current submission to dryad, please use the following link:
<http://datadryad.org/submit?journalID=RSOS&manu=RSOS-190562>

- Competing interests

- Authors' contributions

- Acknowledgements

- Funding statement

Kind regards,

on behalf of Andrew Dunn (Associate Editor) and Kevin Padian (Subject Editor)
openscience@royalsociety.org

Associate Editor's comments (Mr Andrew Dunn):

Comments to the Author:

Please ensure you fully respond to and incorporate the changes suggested by the referees - if you do not satisfy the reviewers that the paper is ready for acceptance, we will be unlikely to consider any further revisions. Good luck!

Reviewers' Comments to Author:

Reviewer: 1

Comments to the Author(s)

Manuscript ID RSOS-190562

Title: Biodegradation of chlorothalonil by strain *Stenotrophomonas acidaminiphila* BJ1 isolated from farmland soil.

The manuscript is well organized and written. The manuscript described the degradation potential, in culture media, of a bacteria strain isolated from a contaminated soil in China to remove chlorothalonil under different concentrations and temperatures. The novelty of this article is scarce since there are several manuscripts published and even the main author has already published the degradation potential of this strain in soil under greenhouse conditions, the enzymes implicated, the degradation pathways as well as genotoxicity of chlorothalonil using micronucleus assay of *V. faba*. It is interesting the study related to identify the isolated strain and biodegradation potential under different the environments. In the experimental design no controls were run in parallel with the experimental solutions. The analysis of the metabolites (formation and elimination) during the degradation of chlorotalonyl in the liquid media is important to describe the degradation pathway. Author only detected two metabolites of the proposed pathway and those were not confirmed. The discussion is a bit speculative. The English grammar was not revised. The article must be improved to be published in this journal. Some more comments.

Introduction

The literature reported must be actualized. There are more recent manuscripts related with the subject of this article, pathways, etc... that are not mentioned.

Page 3 line 33 (14) "significant threats to beneficial micro-and microorganisms, including plants". It mean "macro- and microorganisms"?. Please check

Material and method

Describe the biological assays and insert the reference of the biochemical method applied to identify the strain.

Results

Page 9 section 3.3. If possible describe if the two compounds detected were formed at the same time and if were in the same amount or one was relatively more abundant. It is useful to support the degradation pathway. Author must confirm the identity of these compounds.

Discussion

This section must be improved taking into account the different degradation routes described for chlorothalonil in the literature.

Page 10 lines 183-186 (4-12). This information is a repetition and can be omitted.

Page 10 line 200 (48). "According to our knowledge, this is the first study in which we provide information about the metabolic pathway of chlorothalonil degradation by the *S. acidaminiphila* strain". The compounds were not confirmed by the authors. The pathway is a speculation with data from literature and it does not improve the actual knowledge about the biodegradation pathway of chlorothalonil. This sentence must be take with caution.

Page 12 lines 230-23 (12-15). "Among these two degradation products, the -COOCH₃ group might be esterified in the presence of formic acid during the analysis process. This is unclear because in the material and method section the analysis process was omitted.

I suggest read the following reference to be inserted in the discussion section: Chlorothalonil degradation by *Ochrobactrum lupini* strain TP-D1 and identification of its metabolites by Xiu-Zhen Shi, Rong-Jun Guo, Kazuhiro Takagi, Zuo-Qing Miao, Shi-Dong Li. *World J Microbiol Biotechnol* (2011) 27:1755–1764. DOI 10.1007/s11274-010-0631-0.

Figures

Figure 1. Please increase the size

Figure 6 can be omitted.

Reviewer: 2

Comments to the Author(s)

The work of Zhang et al describes the isolation of a soil bacterium able to biotransform the fungicide chlorothalonil. The authors studied the effect of some conditions (pH, temperature, initial concentration) on the biotransformation of chlorothalonil by this strain and also detected two metabolites. Overall it is a very straight-forward work and I would recommend it for publication after some issues are addressed.

In your 2017 paper (doi: 10.1186/s13568-017-0530-y) you already mention strain BJ1 and have already identified it as *Stenotrophomonas acidaminiphila*, a strain able to utilize chlorothalonil ("The *Stenotrophomonas acidaminiphila* probiotic strain BJ1 was isolated in our laboratory and deposited in China Center for Type Culture Collection"). But this work is presented in the current manuscript as something new. If it is not, then paragraphs 2.2 and 3.1 should be omitted. But if they are indeed omitted, then this manuscript will lack its current completeness.

In the title "biodegradation" could be replaced by "biotransformation".

Line 177: if the bacterium was not able to mineralize the pollutant, then how was it able to grow with the pollutant as a sole carbon and energy source?

Line 69: do you mean 10 g/L NaCl?

Line 72: what was the concentration of the chlorothalonil stock solution? In which solvent it was dissolved?

Line 82: how did you measure the “chlorothalonil-degrading capabilities”?

Line 104: did you use a glass tube? A polypropylene falcon tube?

Line 109: goal was to determine the “optimum conditions” of bacterial growth/ chlorothalonil degradation/both?

Line 155: a calibration curve to correlate the OD with the dry cell weight is needed. Also in line 103 you mention that the initial OD is 1, but here you say that it was 0.03. elaborate?

Line 222: other microorganisms can tolerate higher concentrations of this pollutant fungicide, compared to BJ1?

Line 226: describe the products and pathways.

Figure legends 2 and 4: state the conditions (pH, temp etc)

Figure 3 could be split into two separate figures (3a and 3b) and the time course be presented.

Figure 6: how certain can you be about the position at which the molecule is transformed? (the Cl at position 4 and CN at position 1).

Author's Response to Decision Letter for (RSOS-190562.R0)

See Appendix A.

RSOS-190562.R1 (Revision)

Review form: Reviewer 2

Is the manuscript scientifically sound in its present form?

Yes

Are the interpretations and conclusions justified by the results?

Yes

Is the language acceptable?

Yes

Do you have any ethical concerns with this paper?

No

Have you any concerns about statistical analyses in this paper?

No

Recommendation?

Accept with minor revision (please list in comments)

Comments to the Author(s)

I would like to thank the authors for their response to my comments.

About my comment concerning the tolerance of chlorothalonil by this bacterial strain and other strains. You mention in the discussion that by increasing the initial concentration of the fungicide,

the ability of this strain to biotransform it decreases. So for 500 mg/L the degradation is around 50%, while for 50-300 mg/L degradation is over 80%. Can you compare this with the rest of chlorothalonil-degrading microorganisms? Are other strains able to grow under such high concentrations of chlorothalonil? If so, what is the degradation percentage?

One last thing: the use on the word "rate" (lines 161, 167, 170, 208, 221, 223, 226, 393, 394 and figures) is wrong. Rate incorporates time, but in this case you just mean the difference between the initial and the current concentration (in percentage). If this is divided by time, then it becomes rate.

Decision letter (RSOS-190562.R1)

28-Sep-2019

Dear Dr Zhang:

On behalf of the Editors, I am pleased to inform you that your Manuscript RSOS-190562.R1 entitled "Biotransformation of chlorothalonil by strain *Stenotrophomonas acidaminiphila* BJ1 isolated from farmland soil" has been accepted for publication in Royal Society Open Science subject to minor revision in accordance with the referee suggestions. Please find the referees' comments at the end of this email.

The reviewers and Subject Editor have recommended publication, but also suggest some minor revisions to your manuscript. Therefore, I invite you to respond to the comments and revise your manuscript.

- Ethics statement

- Data accessibility

If you wish to submit your supporting data or code to Dryad (<http://datadryad.org/>), or modify your current submission to dryad, please use the following link:
<http://datadryad.org/submit?journalID=RSOS&manu=RSOS-190562.R1>

- Competing interests

- Authors' contributions

- Acknowledgements

- Funding statement

Because the schedule for publication is very tight, it is a condition of publication that you submit the revised version of your manuscript before 07-Oct-2019. Please note that the revision deadline will expire at 00.00am on this date. If you do not think you will be able to meet this date please let me know immediately.

- 1) A text file of the manuscript (tex, txt, rtf, docx or doc), references, tables (including captions) and figure captions. Do not upload a PDF as your "Main Document".
- 2) A separate electronic file of each figure (EPS or print-quality PDF preferred (either format should be produced directly from original creation package), or original software format)

- 3) Included a 100 word media summary of your paper when requested at submission. Please ensure you have entered correct contact details (email, institution and telephone) in your user account
- 4) Included the raw data to support the claims made in your paper. You can either include your data as electronic supplementary material or upload to a repository and include the relevant doi within your manuscript
- 5) All supplementary materials accompanying an accepted article will be treated as in their final form. Note that the Royal Society will neither edit nor typeset supplementary material and it will be hosted as provided. Please ensure that the supplementary material includes the paper details where possible (authors, article title, journal name).

on behalf of Prof Kevin Padian (Subject Editor)
openscience@royalsociety.org

Reviewer comments to Author:
Reviewer: 2

Comments to the Author(s)
I would like to thank the authors for their response to my comments.

About my comment concerning the tolerance of chlorothalonil by this bacterial strain and other strains. You mention in the discussion that by increasing the initial concentration of the fungicide, the ability of this strain to biotransform it decreases. So for 500 mg/L the degradation is around 50%, while for 50-300 mg/L degradation is over 80%. Can you compare this with the rest of chlorothalonil-degrading microorganisms? Are other strains able to grow under such high concentrations of chlorothalonil? If so, what is the degradation percentage?

One last thing: the use on the word "rate" (lines 161, 167, 170, 208, 221, 223, 226, 393, 394 and figures) is wrong. Rate incorporates time, but in this case you just mean the difference between the initial and the current concentration (in percentage). If this is divided by time, then it becomes rate.

Author's Response to Decision Letter for (RSOS-190562.R1)

See Appendix B.

Decision letter (RSOS-190562.R2)

03-Oct-2019

Dear Dr Zhang,

I am pleased to inform you that your manuscript entitled "Biotransformation of chlorothalonil by strain *Stenotrophomonas acidaminiphila* BJ1 isolated from farmland soil" is now accepted for publication in Royal Society Open Science.

Kind regards,

Lianne Parkhouse
Royal Society Open Science
openscience@royalsociety.org

on behalf of the Associate Editor, and Professor Kevin Padian (Subject Editor)
openscience@royalsociety.org

Appendix A

Dear Dr. Lianne Parkhouse and reviewers,

We deeply appreciate your decision letter on our manuscript titled “Biodegradation of chlorothalonil by strain *Stenotrophomonas acidaminiphila* BJ1 isolated from farmland soil” (RSOS-190562)” on Jul 19, 2019. Thank you very much for providing us the opportunity to revise the manuscript. We also thank the reviewers for providing very constructive comments and suggestions. Overall the comments are fair, encouraging, and helpful to improve the quality of the manuscript.

We have reviewed the manuscript carefully according to all the comments, and revised it **point-by-point** along with a clear indication of the changes in **red color**. We have thoroughly edited the manuscript and now the English of this manuscript is correct.

Thanks for considering the revised version of our manuscript, and we look forward to your reply at your next earliest convenience. Meantime, if you need any other information, please feel free to contact me.

Yours Sincerely,

Qingming Zhang

Qingdao Agricultural University

E-mail: zhangqingminghf@163.com

Adress: No. 700, Changcheng Road, Chengyang District, Qingdao 266109, China.

Response to Reviewer 1

The manuscript is well organized and written. The manuscript described the degradation potential, in culture media, of a bacteria strain isolated from a contaminated soil in China to remove chlorothalonil under different concentrations and temperatures. The novelty of this article is scarce since there are several manuscripts published and even the main author has already published the degradation potential of this strain in soil under greenhouse conditions, the enzymes implicated, the degradation pathways as well as genotoxicity of chlorothalonil using micronucleus assay of *V. faba*. It is interesting the study related to identify the isolated strain and biodegradation potential under different the environments. In the experimental design no controls were run in parallel with the experimental solutions. The analysis of the metabolites (formation and elimination) during the degradation of chlorotalonyl in the liquid media is important to describe the degradation pathway. Author only detected two metabolites of the proposed pathway and those were not confirmed. The discussion is a bit speculative. The English grammar was not revised. The article must be improved to be published in this journal.

Response: Thank you for your general comments. In this study, we mainly want to introduce the degradation characteristics and pathway of the newly isolated degrading bacteria *Stenotrophomonas acidaminiphila* BJ1 on the fungicide chlorothalonil. We think our work is very important to enrich the pesticide degradation microbial resource bank. In our experiment, control experiment was conducted (Lines 109-110). According to your suggestion, the discussion has been rewritten and the English grammar has been corrected.

Some more comments.

Introduction

The literature reported must be actualized. There are more recent manuscripts related with the subject of this article, pathways, etc... that are not mentioned.

Response: Thank you for your advice. We have further supplied some recent manuscripts related with the chlorothalonil degrading microorganisms in the introduction. Lines 50-52.

Page 3 line 33 (14) “significant threats to beneficial micro-and microorganisms, including plants”. It mean “macro- and microorganisms”??. Please check

Response: Thank you for your comments. We have changed this sentence. Line 33.

Material and method

Describe the biological assays and insert the reference of the biochemical method applied to identify the strain.

Response: Thank you for your comments. We have supplied the contents according to your advice. Lines 88-90.

Results

Page 9 section 3.3. If possible describe if the two compounds detected were formed at the same time and if were in the same amount or one was relatively more abundant. It is useful to support the degradation pathway. Author must confirm the identity of these compounds.

Response: Thank you for your comments. Yes, the two compounds were detected at the same time. According to the results of our study, the degradation of chlorothalonil by strain BJ1 existed two degradation pathways. Because we have not bought the standards of two chlorothalonil metabolites, we did not further confirm these compounds in a short time. Based on your suggestion, we will synthesize these compounds to conform our speculation in the future studies. Thank you.

Discussion

This section must be improved taking into account the different degradation routes described for chlorothalonil in the literature.

Response: Thank you for your comments. The published degradation routes of chlorothalonil have been added in the discussion section. Lines 227-231.

Page 10 lines 183-186 (4-12). This information is a repetition and can be omitted.

Response: Thank you for your comments. This section has been deleted.

Page 10 line 200 (48). “According to our knowledge, this is the first study in which we provide information about the metabolic pathway of chlorothalonil degradation by the *S. acidaminiphila* strain”. The compounds were not confirmed by the authors. The pathway is a speculation with data from literature and it does not improve the actual knowledge about the biodegradation pathway of chlorothalonil. This sentence must be take with caution.

Response: Thank you for your comments. We have changed this sentence for avoiding controversy (Lines 201-202).

Page 12 lines 230-23 (12-15). “Among these two degradation products, the -COOCH₃ group might be esterified in the presence of formic acid during the analysis process. This is unclear because in the material and method section the analysis process was omitted.

Response: Thank you for your comments. We have added the analysis process in the material and method section and inserted the relevant reference. Lines 132-133.

I suggest read the following reference to be inserted in the discussion section: Chlorothalonil degradation by *Ochrobactrum lupini* strain TP-D1 and identification of its metabolites by Xiu-Zhen Shi, Rong-Jun Guo, Kazuhiro Takagi, Zuo-Qing Miao, Shi-Dong Li. *World J Microbiol Biotechnol* (2011) 27:1755–1764. DOI 10.1007/s11274-010-0631-0.

Response: Thank you for your comments. This paper you suggested is a high-quality paper. The authors did some wonderful work. The paper you suggested is very useful to improve our paper quality. We have cited it in our paper.

Figures

Figure 1. Please increase the size

Response: Thank you for your advice. The size of Fig.1 has been increased.

Figure 6 can be omitted.

Response: Thank you for your advice. The Fig.6 has been omitted.

Response to Reviewer 2

Comments to the Author(s)

The work of Zhang et al describes the isolation of a soil bacterium able to biotransform the fungicide chlorothalonil. The authors studied the effect of some conditions (pH, temperature, initial concentration) on the biotransformation of chlorothalonil by this strain and also detected two metabolites. Overall it is a very straight-forward work and I would recommend it for publication after some issues are addressed.

Response: Thank you for your positive comments on our work. This will highly encourage us to do better work in the future.

In your 2017 paper (doi: 10.1186/s13568-017-0530-y) you already mention strain BJ1 and have already identified it as *Stenotrophomonas acidaminiphila*, a strain able to utilize chlorothalonil (“The *Stenotrophomonas acidaminiphila* probiotic strain BJ1 was isolated in our laboratory and deposited in China Center for Type Culture Collection”). But this work is presented in the current manuscript as something new. If it is not, then should be omitted. But if they are indeed omitted, then this manuscript will lack its current completeness.

Response: Thank you for your comments. The strain BJ1 have identified and deposited in China Center for Type Culture Collection. But this work (paragraphs 2.2 and 3.1) has not been published in any journals. Moreover, as you’ve said, this manuscript will lack its current completeness if this section be omitted. Therefore, we think this section should not be omitted.

In the title “biodegradation” could be replaced by “biotransformation.

**Response: Thank you for your advice. We have changed “biodegradation” to “biotransformation”.
Line 1.**

Line 177: if the bacterium was not able to mineralize the pollutant, then how was it able to grow with the pollutant as a sole carbon and energy source?

Response: Thank you for your comments. In our study, the strain BJ1 can’t mineralize chlorothalonil. But it can metabolize chlorothalonil into other substances but not CO₂. Previous study also reported that chlorothalonil degrading bacteria TUAH-1 can grow Minimal medium containing chlorothalonil but not mineralize chlorothalonil.

Tang L, Dong J, Ren L, Zhu Q. 2017 Biodegradation of chlorothalonil by *Enterobacter cloacae* TUAH-1. *Int. Biodeter. Biodegr.* 121, 122-130.

Line 69: do you mean 10 g/L NaCl?

Response: Thank you for your comments. The concentration of NaCl is 1.0 g/L. Line 69.

Line 72: what was the concentration of the chlorothalonil stock solution? In which solvent it was dissolved?

Response: Thank you for your comments. In here the concentration of chlorothalonil in stock solution is 1000 mg/L. The solvent is acetone. In our study, for avoiding the possible utilization of acetone as carbon by strain BJ1, the different quantity of chlorothalonil stock solution was firstly added into tube, then the acetone was evaporated before adding the medium. The details have been added in material and methods (Lines 72-75).

Line 82: how did you measure the “chlorothalonil-degrading capabilities”?

Response: Thank you for your comments. We used the method presented in section 2.3.

Line104: did you use a glass tube? A polypropylene falcon tube?

Response: Thank you for your comments. We used glass tubes. This information has been added in our paper (Line 108).

Line 109: goal was to determine the “optimum conditions” of bacterial growth/ chlorothalonil degradation/both?

Response: Thank you for your comments. The optimum conditions mainly refer to biodegradation of chlorothalonil by the strain BJ1.

Line 155: a calibration curve to correlate the OD with the dry cell weight is needed. Also in line 103 you mention that the initial OD is 1, but here you say that it was 0.03. elaborate?

Response: Thank you for your comments. In our study, we firstly made strain BJ1 cell suspension ($OD_{600}=1$), and this cell suspension was added MSM by 5% (v/v). Therefore, the optical density of cells is different in both cases.

Line 222: other microorganisms can tolerate higher concentrations of this pollutant fungicide, compared to BJ1?

Response: Thank you for your comments. We are very sorry for that we did not compare the tolerate ability of other microorganisms on higher concentrations of chlorothalonil with the strain BJ1. We have discussed this point (Lines 223-224).

Line 226: describe the products and pathways.

Response: Thank you for your comments. The products and pathways have been added in here. Lines 227-231.

Figure legends 2 and 4: state the conditions (pH, temp etc)

Response: Thank you for your comments. The conditions have been supplied in figures 2 and 4.

Figure 3 could be split into two separate figures (3a and 3b) and the time course be presented.

Response: Thank you for your comments. Fig. 3 has been split into two figures. The results were

obtained at one sample time. Therefore, the time course has not been presented in the Fig.3.

Figure 6: how certain can you be about the position at which the molecule is transformed? (the Cl at position 4 and CN at position 1).

Response: Thank you for your comments. In here, the position of Cl and CN in benzene ring was speculated based on our result and previous published relevant papers. According to the suggestion of another reviewer, the fig.6 has been omitted in this paper.

Appendix B

Comments to the Author(s)

I would like to thank the authors for their response to my comments.

About my comment concerning the tolerance of chlorothalonil by this bacterial strain and other strains. You mention in the discussion that by increasing the initial concentration of the fungicide, the ability of this strain to biotransform it decreases. So for 500 mg/L the degradation is around 50%, while for 50-300 mg/L degradation is over 80%. Can you compare this with the rest of chlorothalonil-degrading microorganisms? Are other strains able to grow under such high concentrations of chlorothalonil? If so, what is the degradation percentage?

Response: Thank you for your comments. According to your suggestion, the question you pointed out has been discussed in this manuscript. Please see lines 223-230.

One last thing: the use on the word "rate" (lines 161, 167, 170, 208, 221, 223, 226, 393, 394 and figures) is wrong. Rate incorporates time, but in this case you just mean the difference between the initial and the current concentration (in percentage). If this is divided by time, then it becomes rate.

Response : Thank you for pointing out the errors in our manuscript. The “degradation rate” in lines of 161, 167, 170, 208, 221, 223, 226, 393, 394 and figures have been corrected to “degradation efficiency”.